# Influence of Sex and Dominant Side on the Reliability of Two Trunk Rotator Exercises

Angela Rodríguez-Perea [1,2], Daniel Jerez-Mayorga [1,2,3,*], María Dolores Morenas-Aguilar [1,2], Darío Martínez-García [1,2], Ignacio Jesús Chirosa-Ríos [1,2], Luis Javier Chirosa-Ríos [1,2] and Waleska Reyes-Ferrada [3]

1   Department of Physical Education and Sport, Faculty of Sports Sciences, University of Granada, 18071 Granada, Spain
2   CTS-642 Research Group, Strength & Conditioning Laboratory, Department Physical Education and Sports, Faculty of Sport Sciences, University of Granada, 18071 Granada, Spain
3   Exercise and Rehabilitation Sciences Laboratory, Faculty of Rehabilitation Sciences, School of Physical Therapy, Universidad Andres Bello, Santiago 7591538, Chile
*   Correspondence: djerezmayorga@ugr.es

**Abstract:** Background: A method to assess the influence of sex and side testing on trunk rotator muscles has not been described. The purpose was to analyze the influence of sex and dominant and non-dominant sides (DS-NDS) on the reliability of two trunk rotator exercises and to study the relationship between the DS-NDS of two trunk rotator strength exercises. Methods: The reliability of the horizontal cable woodchop (HCW) and low cable woodchop (LWC) exercises was studied using a test-retest design with 51 physically active students. Isokinetic and isometric strength were assessed with a functional electromechanical dynamometer. Results: There were significant differences in reliability between male and female HCW and no significant differences in reliability between the average of the DS-NDS in HCW and LCW. There were no significant differences between the DS-NDS in the sex of HCW, and the strength of the two exercises showed no significant differences except for two conditions assessed. Very large to extremely large correlations were observed between sides in the strength of two exercises (r = 0.71–0.91). Conclusions: This test is handy for physical trainers or coaches to know the strength of the trunk rotators of their athletes.

**Keywords:** muscle strength dynamometer; reproducibility; isokinetic

## 1. Introduction

Muscle imbalance seems to be present in unilateral sports such as golf, handball, or volleyball [1–3], and it is an important indicator of the risk of injury [4,5]. Especially in the trunk, a muscle imbalance and decreased strength of the extensor muscles are closely related to the occurrence of non-specific low back pain, and it's used to predict injury due to the multiple methods to evaluate it [6].

On the one hand, different authors have assessed the muscle performance of trunk musculature to improve performance and reduce the risk of injury in rotational sports. In unilateral sports, the difference in training volume between the dominant side and non-dominant side (DS-NDS) depends on the amount of time spent on specific field training and conditioning training [7]. The more hours of specific training without compensating strategies for the less-trained limb, the greater the muscle imbalance and the risk of musculoskeletal injury for athletes. It has been proved that golfers with low back pain have reduced muscle endurance on the non-dominant side compared to healthy subjects [8]. Furthermore, it has been shown that trunk strength is critical in sports such as judo [9]. The strength of the trunk rotator musculature (STRM) for the prevention of injuries in athletes and for improving performance has been shown to play a fundamental role; therefore, it is

necessary to create reliable tests to find out the improvements in strength in this musculature and to study the different variables that can affect the evaluation, such as sex, the velocity of the exercise, or the direction of the force. There is controversy in the literature as to whether the reliability of STRM assessment is influenced by sex, with differences in reliability in one study [10] and similarities in another [11].

Until the last few years, the STRM has been evaluated with isokinetic devices in isolation or isometric strength with hand-held dynamometers [12,13]. However, this mode of physical evaluation is far from sport-specific performance and common daily activities. Nowadays, with the development of elite sports, the main purpose of strength and conditioning coaches and rehabilitators is to assess strength in the most similar way to sporting gestures. In this way, different studies have evaluated the STRM in a similar way to sporting gestures, using a system of pulleys and dynamometers [14–16] or load cells [17]. However, there has been a growth in sports science technology associated with new motorized electrical devices that allow the assessment of strength in different modes such as isokinetic, tonic, isoinertial, eccentric, elastic, and isometric (Dynasystem, Exentrix, Quantum 1080, Globus Lineo, MyoQuality) [18,19]. These devices can solve the necessity of functional assessment; however, the first step is to develop reliable tests to assess human strength.

For all of the above reasons, there is a growing necessity to create specific and reliable tests to evaluate the STRM. Therefore, the objectives were to analyze the influence of sex on the reliability of two trunk rotator exercises (TRE), compare the reliability of the DS-NDS in the assessment of two TRE, and study the relationship between the DS-NDS of two TRE strengths. We hypothesized that sex does not affect the reliability of TRE and that the dominant side is more reliable than the non-dominant side. Another hypothesis was that the relationship between the DS-NDS in assessing of two TRE is very large.

## 2. Materials and Methods

### 2.1. Participants

Twenty-two females (20.90 $\pm$ 2.30 years; 65.55 $\pm$ 9.65 kg; 1.64 $\pm$ 0.06 m; and 24.26 $\pm$ 3.59 kg/m$^2$) and thirty males (23.43 $\pm$ 3.88 years; 78.23 $\pm$ 12.21 kg; 1.76 $\pm$ 0.06 m; and 25.00 $\pm$ 3.11 kg/m$^2$) are physically active college students engaged in training sessions at least three times a week, with a minimum of 6 months of structured resistance training and without any musculoskeletal injuries. The evaluations were conducted by three sports scientists with more than eight years of experience performing muscle strength tests and using the device. (AR-P, WR-F, and MDM-A). Participants with an index greater than 20% in the Oswestry Disability Index (ODI), a history of neurological or cardiorespiratory pathology, or abdominal surgeries within the last 6 months, and who performed specific trunk exercises, were excluded from the study. Furthermore, they were informed about the nature, aims, and risks associated with the experimental procedure before giving their written consent to participate. The study protocol was approved by the Institutional Review Board of the University of Granada (n° 2560/CEIH/2022) and was conducted following the Helsinki Declaration.

### 2.2. Study Design

A test-retest design was performed to analyze the influence of sex and side dominance on the reliability of the horizontal cable woodchop (HCW) and low cable woodchop (LWC) exercises. After a familiarization session, the subjects attended the laboratory for two days separated by 48 h. The participants executed two exercises with different protocols for assessing the STRM in isokinetic mode and isometric mode.

### 2.3. Testing Procedures

Isokinetic and isometric strength were evaluated with a functional electromechanical dynamometer (FEMD) (Dynasystem, Symotech, Granada, Spain) [18]. A standard grip was used to execute the two TRE. The distance between the acromion and the knuckle of the

middle finger was measured manually to establish the range of motion (ROM) of the HCW exercise, and the distance between the acromion and the end of the middle finger was the ROM of the LCW exercise. Measurements were made by applying an anthropometric measurement protocol based on the internationally validated recommendation [20] and using a SECA brand measuring tape.

The initial position of HCW was with the participant standing with feet shoulder-width apart, holding the grip with both hands, elbows extended, and the shoulder flexed at 90º measured with a goniometer (Gymna hoofdzetel, Bilzen, Belgium). Then, with a slight flexion of the knees (<20°), the participant rotated the trunk with a forceful horizontal movement until the end of the ROM (concentric phase) and then had to try to retain the movement back (eccentric phase) (Figure 1).

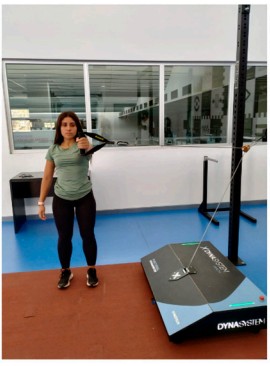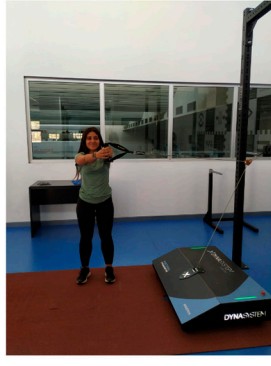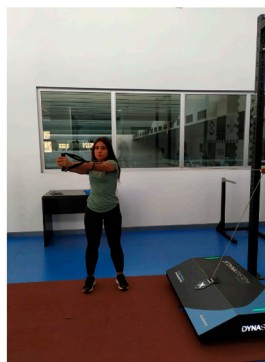

**Figure 1.** Assessment of horizontal cable woodchop exercise using a functional electromechanical dynamometer.

And the initial position of the LCW exercise was with the participant standing with feet shoulder-width apart, holding the grip with both hands below the hip, (at the level of the anterior superior iliac spine), elbows extended. The initial position of the arms was with the grip in contact with the participant's thigh. Then, with a slight flexion of the knees (<20°), the participant rotated the trunk with a forceful movement while raising the arms diagonally until the end of the ROM (concentric phase) and then had to try to retain the movement of return (eccentric phase) (Figure 2).

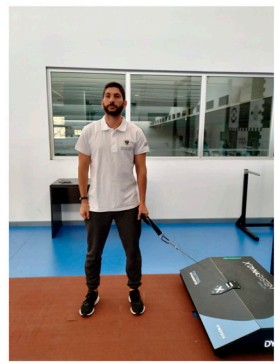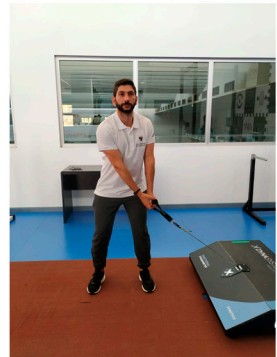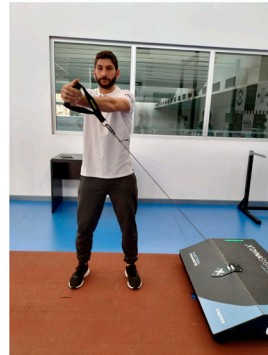

**Figure 2.** Assessment of low cable woodchop exercise using a functional electromechanical dynamometer.

The participants attended a familiarization session with the FEMD. Familiarization consisted of a 60-min session, a 10-min warm-up, and then the familiarization with the FEMD was performed at the same velocities as the assessment, with a specific ROM for each subject (one series of two submaximal and five maximal repetitions). These velocities were selected for participants to ensure that they did not lose the correct exercise technique. In this session, the ROM for each subject was recorded. The instructions given to the participants were always the same, and no feedback or encouragement was provided.

After the warm-up, participants performed one set of five repetitions of two exercises in each test condition. The LCW and HCW exercises were tested in isokinetic mode at two different speeds ($0.50 \, \text{m} \cdot \text{s}^{-1}$ and $0.70 \, \text{m} \cdot \text{s}^{-1}$) and ($0.40 \, \text{m} \cdot \text{s}^{-1}$ and $0.60 \, \text{m} \cdot \text{s}^{-1}$), respectively. In addition, a 6-s isometric evaluation was performed in the initial position of each exercise. Between each series, rest for 3 min. The rater was blind to the results of their measurements and additional cues that were not part of the test.

### 2.4. Statistical Analysis

The three highest repetitions of the average and the peak strength were taken to calculate the dynamic strength. In calculating the isometric strength, the repetition's peak and mean values were taken. Retest data were used to study the relationship between the strength of the DS-NDS in the two TREs.

The descriptive data are presented as a mean $\pm$ SD. The Shapiro–Wilk normality test verified the data distribution. Reliability was assessed by t-tests of paired samples with the effect size (ES), the coefficient of variation (CV), the standard error of measurement (SEM), and the intraclass correlation coefficient (ICC), with 95% confidence intervals (CI). Absolute reliability was assessed using the CV and SEM, while relative reliability was assessed using the ICC (model 3.1) with their respective 95% CIs. The scale used for interpreting the magnitude of the ES was specific to training research [21], and the magnitude of the values of the intraclass correlation coefficient was measured on a qualitative scale [22]. To interpret the observed magnitude of differences in coefficients of variation, a default for the smallest important ratio of 1.15 was used [23–25]. For the CVratio study, the mean values of dominant and non-dominant sides in all assessment conditions were calculated to compare the reliability of males and females. Furthermore, the mean values of males and females were calculated to compare the reliability of the dominant and non-dominant sides.

A paired-sample t-test was used to determine the differences between the STRM of DS-NDS on two exercises. The level of significance was set at $p < 0.05$. Also, a Pearson correlation coefficient with a 95% CI was calculated for the relation between the strength of the TRM of the DS-NDS on two exercises. The criteria to interpret the magnitude of the r were small (0.10–0.29), moderate (0.30–0.49), large (0.50–0.69), very large (0.70–0.89), and extremely large (0.90–1.00). Reliability analyses were performed using a customized spreadsheet [26] and the SPSS software package (version 25.0).

## 3. Results

Fifty-one subjects participated in the study. Unfortunately, a female dropped out of the study due to a musculoskeletal injury. The average and peak strength values (mean (SD)) of the two TRE are shown in Tables 1 and 2. The absolute and relative reliability values of the two sexes and DS-NDS are shown in Tables 3 and 4.

There were significant differences between males and females during the HCW exercise. When the average (CVratio = 1.42) and peak strength (CVratio = 1.23) were taken, the female showed to be more reliable (CV = 9.97% vs. CV = 14.16%; CV = 9.84% vs. CV = 13.01% respectively). However, in the LCW exercise, there were no significant differences when taking the average or peak strength with a CVratio > 1.15.

**Table 1.** Absolute and relative test-retest reliability of the average and peak strength low cable woodchop exercise.

| | | Test (Kg) | Retest (Kg) | *p*-Value | ES | ICC (95% CI) | CV (95% CI) | SEM (95% CI) |
|---|---|---|---|---|---|---|---|---|
| **Average Strength** | | | | | | | | |
| LCW 0.50 m·s⁻¹ | Con ND | 14.9 (5.1) | 14.4 (4.7) | 0.22 | −0.09 | 0.86 (0.77–0.92) | 12.58 (10.53–15.64) | 1.84 (1.54–2.29) |
| | Ecc ND | 23.8 (6.9) | 22.6 (6.7) | 0.04 | −0.18 | 0.83 (0.72–0.90) | 12.28 (10.28–15.27) | 2.85 (2.38–3.54) |
| | Con D | 15.9 (6.0) | 15.0 (4.4) | 0.04 | −0.16 | 0.85 (0.75–0.91) | 13.32 (11.14–16.56) | 2.06 (1.72–2.56) |
| | Ecc D | 23.0 (7.8) | 22.7 (5.6) | 0.64 | −0.05 | 0.76 (0.62–0.86) | 14.73 (12.32–18.30) | 3.37 (2.82–4.19) |
| LCW 0.70 m·s⁻¹ | Con ND | 19.0 (8.6) | 14.2 (4.5) | 0.01 | −0.69 | 0.58 (0.36–0.74) | 26.97 (22.53–33.61) | 4.48 (3.74–5.58) |
| | Ecc ND | 21.2 (5.9) | 23.3 (6.4) | 0.01 | 0.32 | 0.70 (0.53–0.82) | 15.47 (12.92–19.28) | 3.43 (2.87–4.28) |
| | Con D | 15.5 (5.4) | 14.7 (4.9) | 0.03 | −0.15 | 0.89 (0.82–0.94) | 11.32 (9.47–14.08) | 1.71 (1.43–2.12) |
| | Ecc D | 24.6 (6.3) | 23.0 (6.3) | 0.01 | −0.25 | 0.81 (0.69–0.89) | 11.69 (9.78–14.53) | 2.78 (2.32–3.45) |
| LCW Iso | Iso ND | 21.7 (8.3) | 22.8 (6.5) | 0.06 | 0.15 | 0.85 (0.75–0.91) | 13.33 (11.16–16.58) | 2.97 (2.48–3.69) |
| | Iso D | 21.2 (7.1) | 22.7 (7.2) | 0.03 | 0.21 | 0.78 (0.65–0.87) | 15.31 (12.81–19.03) | 3.36 (2.81–4.18) |
| **Peak Strength** | | | | | | | | |
| LCW 0.50 m·s⁻¹ | Con ND | 24.7 (9.2) | 24.9 (9.1) | 0.86 | 0.01 | 0.82 (0.71–0.89) | 15.87 (13.28–19.72) | 3.94 (3.29–4.89) |
| | Ecc ND | 39.6 (12.3) | 37.2 (11.4) | 0.05 | −0.21 | 0.74 (0.59–0.84) | 15.91 (13.31–19.77) | 6.10 (5.11–7.59) |
| | Con D | 25.1 (8.9) | 25.1 (9.0) | 0.93 | −0.01 | 0.78 (0.64–0.87) | 17.07 (14.28–21.22) | 4.28 (3.58–5.33) |
| | Ecc D | 40.4 (15.4) | 38.4 (12.3) | 0.26 | −0.14 | 0.62 (0.41–0.76) | 22.14 (18.52–27.52) | 8.72 (7.30–10.84) |
| LCW 0.70 m·s⁻¹ | Con ND | 25.1 (9.7) | 24.8 (11.0) | 0.63 | −0.03 | 0.89 (0.81–0.93) | 14.32 (11.96–17.84) | 3.57 (2.98–4.45) |
| | Ecc ND | 43.1 (14.0) | 40.2 (14.1) | 0.02 | −0.20 | 0.82 (0.71–0.90) | 14.42 (12.04–17.97) | 6.00 (5.01–7.48) |
| | Con D | 26.2 (10.7) | 24.0 (8.3) | 0.02 | −0.24 | 0.76 (0.62–0.86) | 18.96 (15.86–23.57) | 4.76 (3.99–2.92) |
| | Ecc D | 44.8 (16.1) | 40.5 (13.3) | 0.01 | −0.29 | 0.75 (0.60–0.85) | 17.64 (14.76–21.93) | 7.53 (6.30–9.36) |
| LCW Iso | Iso ND | 25.8 (10.6) | 26.6 (8.2) | 0.34 | 0.09 | 0.79 (0.66–0.88) | 16.73 (14.00–20.79) | 4.39 (3.67–5.45) |
| | Iso D | 25.0 (9.9) | 26.1 (8.7) | 0.25 | 0.12 | 0.72 (0.55–0.85) | 19.67 (16.45–24.45) | 5.03 (4.21–6.25) |

LWC = low cable woodchop; ISO = isometric contraction; CON = concentric contraction; ECC = eccentric contraction; ND= non-dominant; D = dominant; CV = coefficient of variation; SEM = standard error of measurement (kg); ICC = intraclass correlation coefficient; 95% CI = 95% confidence interval.

**Table 2.** Absolute and relative test-retest reliability of the average and peak strength horizontal cable woodchop exercise.

| | | Test (Kg) | Retest (Kg) | *p*-Value | ES | ICC (95% CI) | CV (95% CI) | SEM (95% CI) |
|---|---|---|---|---|---|---|---|---|
| Average Strength | | | | | | | | |
| HCW 0.40 m·s$^{-1}$ | Con ND | 8.6 (2.4) | 8.4 (2.0) | 0.30 | −0.11 | 0.73 (0.57–0.84) | 13.84 (11.58–17.21) | 1.18 (0.99–1.46) |
| | Ecc ND | 14.6 (3.4) | 14.4 (3.1) | 0.52 | −0.06 | 0.80 (0.68–0.88) | 10.27 (8.59–12.77) | 1.49 (1.24–1.85) |
| | Con D | 9.1 (2.8) | 9.2 (2.9) | 0.57 | 0.04 | 0.89 (0.82–0.94) | 10.21 (8.54–12.69) | 0.94 (0.78–1.16) |
| | Ecc D | 14.8 (3.3) | 14.8 (3.6) | 0.95 | 0.00 | 0.85 (0.75–0.91) | 9.33 (7.80–11.59) | 1.38 (1.15–1.71) |
| HCW 0.60 m·s$^{-1}$ | Con ND | 8.3 (2.3) | 8.1 (2.2) | 0.62 | −0.06 | 0.66 (0.47–0.79) | 15.93 (13.33–19.80) | 1.31 (1.09–1.63) |
| | Ecc ND | 15.1 (3.4) | 14.5 (3.1) | 0.08 | −0.19 | 0.73 (0.58–0.84) | 11.51 (9.63–14.30) | 1.71 (1.43–2.12) |
| | Con D | 8.9 (3.0) | 8.9 (2.9) | 0.89 | −0.01 | 0.86 (0.77–0.92) | 12.33 (10.32–15.33) | 1.10 (0.92–1.37) |
| | Ecc D | 15.1 (3.5) | 15.0 (3.5) | 0.71 | −0.03 | 0.79 (0.66–0.88) | 10.78 (9.02–13.40) | 1.62 (1.36–2.02) |
| HCW Iso | Iso ND | 9.6 (2.7) | 9.3 (2.2) | 0.34 | −0.12 | 0.64 (0.44–0.78) | 15.67 (13.11–19.48) | 1.48 (1.24–1.83) |
| | Iso D | 9.7 (3.9) | 9.5 (2.2) | 0.69 | −0.05 | 0.54 (0.31–0.71) | 22.46 (18.79–27.92) | 2.16 (1.81–2.69) |
| Peak Strength | | | | | | | | |
| HCW 0.40 m·s$^{-1}$ | Con ND | 14.3 (3.5) | 14.6 (3.1) | 0.45 | 0.09 | 0.69 (0.51–0.81) | 12.96 (10.84–16.10) | 1.87 (1.56–2.32) |
| | Ecc ND | 19.7 (3.9) | 19.3 (4.0) | 0.25 | −0.10 | 0.82 (0.70–0.89) | 8.71 (7.28–10.82) | 1.70 (1.42–2.11) |
| | Con D | 14.7 (3.2) | 14.6 (3.4) | 0.94 | −0.01 | 0.64 (0.44–0.78) | 13.66 (11.43–16.98) | 2.00 (1.68–2.49) |
| | Ecc D | 19.4 (3.9) | 19.7 (4.3) | 0.62 | 0.05 | 0.74 (0.58–0.84) | 10.84 (9.07–13.48) | 2.12 (1.77–2.63) |
| HCW 0.60 m·s$^{-1}$ | Con ND | 16.0 (3.8) | 15.8 (3.6) | 0.71 | −0.05 | 0.56 (0.34–0.72) | 15.66 (13.11–19.47) | 2.48 (2.08–3.09) |
| | Ecc ND | 21.2 (4.3) | 20.7 (4.4) | 0.29 | −0.10 | 0.78 (0.65–0.87) | 9.71 (8.13–12.07) | 2.03 (1.70–2.53) |
| | Con D | 16.1 (3.8) | 16.1 (3.8) | 1.00 | 0.00 | 0.71 (0.54–0.82) | 12.90 (10.79–16.03) | 2.07 (1.74–2.58) |
| | Ecc D | 21.5 (5.0) | 20.8 (4.5) | 0.15 | −0.14 | 0.78 (0.65–0.87) | 10.66 (8.92–13.26) | 2.26 (1.89–2.81) |
| HCW Iso | Iso ND | 10.9 (3.3) | 10.5 (2.5) | 0.26 | −0.14 | 0.60 (0.40–0.75) | 17.51 (14.65–21.76) | 1.87 (1.57–2.33) |
| | Iso D | 10.6 (3.0) | 10.7 (2.7) | 0.75 | 0.03 | 0.80 (0.68–0.88) | 12.06 (10.09–14.99) | 1.28 (1.07–1.60) |

HCW = horizontal cable woodchop; ISO = isometric contraction; CON = concentric contraction; ECC = eccentric contraction; ND = non-dominant; D = dominant; CV = coefficient of variation; SEM = standard error of measurement (Kg); ICC = intraclass correlation coefficient; 95% CI = 95% confidence interval.

**Table 3.** Women and men reliability of average and peak strength low cable woodchop exercise.

| | | Woman | | | Men | | |
|---|---|---|---|---|---|---|---|
| | | ICC (95% CI) | CV (95% CI) | SEM (95% CI) | ICC (95% CI) | CV (95% CI) | SEM (95% CI) |
| Average Strength | | | | | | | |
| LCW 0.50 m·s⁻¹ | Con ND | 0.74 (0.46–0.89) | 13.06 (9.99–18.85) | 1.44 (1.10–2.08) | 0.80 (0.61–0.90) | 12.22 (9.74–16.43) | 2.10 (1.67–2.83) |
| | Ecc ND | 0.48 (0.08–0.75) | 14.75 (11.28–21.30) | 2.66 (2.03–3.84) | 0.79 (0.60–0.89) | 10.93 (8.71–14.70) | 2.93 (2.33–3.94) |
| | Con D | 0.86 (0.68–0.94) | 9.14 (6.99–13.20) | 1.08 (0.82–1.55) | 0.78 (0.58–0.89) | 13.46 (10.72–18.10) | 2.43 (1.93–3.27) |
| | Ecc D | 0.77 (0.52–0.90) | 9.49 (7.26–13.71) | 1.75 (1.34–2.52) | 0.67 (0.41–0.83) | 15.27 (12.16–20.52) | 3.97 (3.16–5.34) |
| LCW 0.70 m·s⁻¹ | Con ND | 0.34 (−0.11–0.67) | 21.84 (16.61–31.90) | 2.70 (2.05–3.95) | 0.47 (0.14–0.71) | 27.07 (21.56–36.40) | 5.25 (4.18–7.06) |
| | Ecc ND | 0.47 (0.04–0.75) | 18.63 (14.17–27.22) | 3.39 (2.58–4.95) | 0.66 (0.40–0.82) | 13.61 (10.84–18.30) | 3.38 (2.69–4.55) |
| | Con D | 0.81 (0.58–0.92) | 11.01 (8.42–15.89) | 1.27 (0.97–1.83) | 0.85 (0.71–0.93) | 11.27 (8.97–15.15) | 1.98 (1.58–2.66) |
| | Ecc D | 0.66 (0.32–0.84) | 13.35 (10.22–19.28) | 2.67 (2.04–3.85) | 0.79 (0.61–0.89) | 10.95 (8.72–14.72) | 2.89 (2.30–3.89) |
| LCW Iso | Iso ND | 0.63 (0.29–0.83) | 15.80 (12.09–22.81) | 2.77 (2.12–4.00) | 0.85 (0.70–0.92) | 11.63 (9.26–15.64) | 2.97 (2.37–4.00) |
| | Iso D | 0.74 (0.36–0.89) | 13.18 (9.55–21.23) | 2.38 (1.72–3.83) | 0.77 (0.57–0.88) | 14.68 (11.69–19.74) | 3.63 (2.89–4.88) |
| Peak Strength | | | | | | | |
| LCW 0.50 m·s⁻¹ | Con ND | 0.75 (0.47–0.89) | 16.49 (12.61–23.81) | 3.19 (2.44–4.61) | 0.77 (0.58–0.89) | 15.38 (12.25–20.68) | 4.40 (3.51–5.92) |
| | Ecc ND | 0.49 (0.08–0.76) | 17.87 (13.67–25.81) | 5.47 (4.19–7.90) | 0.68 (0.43–0.83) | 14.96 (11.92–20.11) | 6.56 (5.22–8.81) |
| | Con D | 0.64 (0.29–0.83) | 19.69 (15.06–28.43) | 3.90 (2.99–5.64) | 0.74 (0.52–0.87) | 15.86 (12.63–21.32) | 4.57 (3.64–6.14) |
| | Ecc D | 0.68 (0.36–0.85) | 17.27 (13.21–24.93) | 5.55 (4.25–8.01) | 0.49 (0.17–0.72) | 23.30 (18.56–31.33) | 10.36 (8.25–13.93) |
| LCW 0.70 m·s⁻¹ | Con ND | 0.72 (0.41–0.88) | 15.77 (12.00–23.04) | 2.96 (2.25–4.32) | 0.88 (0.76–0.94) | 13.52 (10.77–18.17) | 3.93 (3.13–5.28) |
| | Ecc ND | 0.65 (0.30–0.84) | 17.07 (12.98–24.93) | 5.72 (4.35–8.36) | 0.82 (0.66–0.91) | 13.19 (10.50–17.73) | 6.20 (4.94–8.33) |
| | Con D | 0.80 (0.56–0.91) | 16.42 (12.56–23.72) | 3.26 (2.50–4.71) | 0.68 (0.43–0.83) | 19.11 (15.22–25.69) | 5.50 (4.38–7.40) |
| | Ecc D | 0.75 (0.47–0.89) | 16.24 (12.42–23.44) | 5.70 (4.36–8.23) | 0.69 (0.44–0.84) | 17.77 (14.15–23.88) | 8.53 (6.79–11.47) |
| LCW Iso | Iso ND | 0.52 (0.12–0.77) | 19.57 (14.97–28.27) | 3.97 (3.04–5.73) | 0.79 (0.61–0.90) | 14.37 (11.44–19.32) | 4.37 (3.48–5.87) |
| | Iso D | 0.72 (0.31–0.88) | 13.64 (9.89–21.98) | 2.77 (2.01–4.47) | 0.78 (0.47–0.89) | 16.31 (12.05–25.24) | 4.77 (3.52–7.38) |

LWC = low cable woodchop; ISO = isometric contraction; CON = concentric contraction; ECC = eccentric contraction; ND= non-dominant; D = dominant; CV = coefficient of variation; SEM = standard error of measurement (kg); ICC = intraclass correlation coefficient; 95% CI = 95% confidence interval.

**Table 4.** Women and men reliability of average and peak strength horizontal cable woodchop exercise.

| | | Woman | | | Men | | |
|---|---|---|---|---|---|---|---|
| | | ICC (95% CI) | CV (95% CI) | SEM (95% CI) | ICC (95% CI) | CV (95% CI) | SEM (95% CI) |
| **Average Strength** | | | | | | | |
| HCW 0.40 m·s$^{-1}$ | ConND | 0.63 (0.28–0.83) | 13.72 (10.50–19.82) | 0.98 (0.75–1.42) | 0.65 (0.39–0.82) | 13.88 (11.05–18.65) | 1.31 (1.04–1.76) |
| | Ecc ND | 0.80 (0.57–0.91) | 9.35 (7.15–13.50) | 1.20 (0.92–1.73) | 0.75 (0.54–0.87) | 10.71 (8.53–14.39) | 1.68 (1.33–2.25) |
| | Con D | 0.86 (0.68–0.94) | 10.16 (7.77–14.67) | 0.79 (0.60–1.14) | 0.88 (0.77–0.94) | 10.05 (8.01–13.52) | 1.02 (0.81–1.38) |
| | Ecc D | 0.80 (0.58–0.92) | 10.02 (7.67–14.48) | 1.31 (1.00–1.89) | 0.83 (0.67–0.92) | 9.05 (7.21–12.17) | 1.44 (1.15–1.94) |
| HCW 0.60 m·s$^{-1}$ | Con ND | 0.76 (0.49–0.89) | 11.20 (8.57–16.18) | 0.76 (0.58–1.09) | 0.43 (0.08–0.68) | 17.32 (13.79–23.28) | 1.60 (1.27–2.15) |
| | Ecc ND | 0.92 (0.80–0.97) | 5.88 (4.49–8.48) | 0.78 (0.59–1.12) | 0.58 (0.29–0.78) | 13.43 (10.69–18.05) | 2.14 (1.71–2.88) |
| | Con D | 0.88 (0.73–0.95) | 10.80 (8.26–15.60) | 0.79 (0.60–1.14) | 0.81 (0.64–0.91) | 12.75 (10.16–17.14) | 1.29 (1.02–1.73) |
| | Ecc D | 0.84 (0.65–0.93) | 9.34 (7.14–13.48) | 1.22 (0.94–1.77) | 0.69 (0.44–0.84) | 11.37 (9.05–15.28) | 1.86 (1.48–2.50) |
| HCW Iso | Iso ND | 0.78 (0.54–0.91) | 10.00 (7.65–14.44) | 0.80 (0.61–1.16) | 0.46 (0.13–0.70) | 17.35 (13.82–23.33) | 1.80 (1.44–2.42) |
| | Iso D | 0.83 (0.63–0.93) | 9.24 (7.07–13.34) | 0.77 (0.59–1.11) | 0.44 (0.10–0.69) | 25.73 (20.49–34.59) | 2.71 (2.16–3.64) |
| **Peak Strength** | | | | | | | |
| HCW 0.40 m·s$^{-1}$ | Con ND | 0.27 (−0.17–0.62) | 14.60 (11.17–21.09) | 1.79 (1.37–2.58) | 0.63 (0.35–0.80) | 12.24 (9.75–16.45) | 1.95 (1.56–2.63) |
| | Ecc ND | 0.71 (0.41–0.87) | 9.34 (7.14–13.48) | 1.59 (1.22–2.30) | 0.78 (0.59–0.89) | 8.22 (6.54–11.05) | 1.74 (1.39–2.34) |
| | Con D | 0.47 (0.06–0.74) | 13.51 (10.34–19.51) | 1.70 (1.30–2.45) | 0.52 (0.20–0.74) | 13.53 (10.78–18.19) | 2.18 (1.74–2.93) |
| | Ecc D | 0.86 (0.68–0.94) | 7.18 (5.50–10.38) | 1.23 (0.94–1.77) | 0.56 (0.25–0.76) | 12.04 (9.59–16.18) | 2.56 (2.04–3.44) |
| HCW 0.60 m·s$^{-1}$ | Con ND | 0.66 (0.33–0.85) | 11.52 (8.81–16.63) | 1.58 (1.21–2.28) | 0.21 (−0.16–0.52) | 16.71 (13.31–22.46) | 2.96 (2.36–3.98) |
| | Ecc ND | 0.89 (0.75–0.95) | 5.22 (3.99–7.54) | 0.95 (0.72–1.37) | 0.64 (0.37–0.81) | 11.08 (8.82–14.89) | 2.54 (2.02–3.41) |
| | Con D | 0.60 (0.23–0.81) | 13.60 (10.41–19.64) | 1.82 (1.40–2.63) | 0.53 (0.21–0.74) | 12.59 (10.03–16.93) | 2.26 (1.80–3.04) |
| | Ecc D | 0.88 (0.74–0.95) | 6.50 (4.97–9.39) | 1.19 (0.91–1.72) | 0.64 (0.37–0.81) | 12.05 (9.60–16.20) | 2.79 (2.23–3.76) |
| HCW Iso | Iso ND | 0.76 (0.49–0.89) | 10.50 (8.04–15.17) | 0.93 (0.71–1.35) | 0.40 (0.05–0.66) | 19.20 (15.29–25.81) | 2.30 (1.83–3.09) |
| | Iso D | 0.91 (0.77–0.96) | 6.38 (4.62–10.27) | 0.58 (0.42–0.94) | 0.77 (0.57–0.88) | 12.47 (9.93–16.77) | 1.46 (1.16–1.96) |

HCW = horizontal cable woodchop; ISO = isometric contraction; CON = concentric contraction; ECC = eccentric contraction; ND = non-dominant; D = dominant; CV = coefficient of variation; SEM = standard error of measurement (Kg); ICC = intraclass correlation coefficient; 95% CI = 95% confidence interval.

There were no significant differences in reliability between DS-NDS (CVratio < 1.15) when the average or peak strength in the HCW exercise was taken. However, in the LCW exercise, there were significant differences in the reliability, with the dominant side being more reliable when the average strength was taken (CVratio = 1.21) and the non-dominant sides when the peak strength was taken (CVratio = 1.24). The only evaluation condition where there were no significant differences between DS-NDS, regardless of the manifestation of the strength taken, was in the concentric phase at a velocity of $0.50 \text{ m s}^{-1}$ (CVratio = 1.05) and $0.70 \text{ m s}^{-1}$ (CVratio = 1.07). Analyzing reliability by sex, no differences were found between DS-NDS in female and male (CVratio < 1.15) in HCW, but there were significant differences in female in the LCW exercise when the average strength was taken (CVratio = 1.49), the dominant side being more reliable, and in male when the peak strength was taken (CVratio = 1.29), the non-dominant side being more reliable (Figure 3).

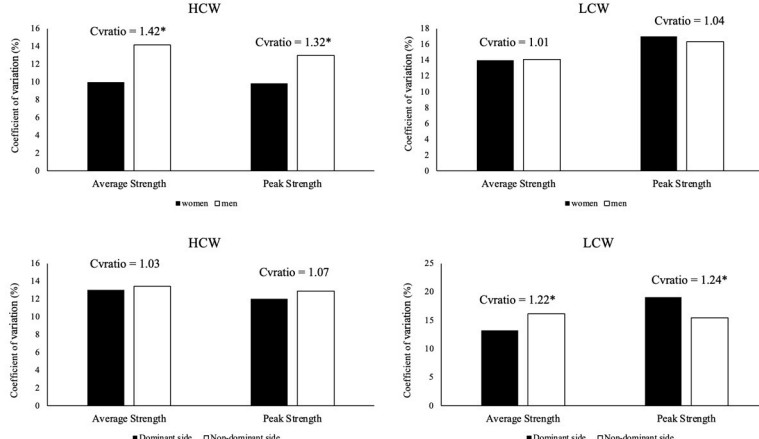

**Figure 3.** Comparison of the reliability of the outcomes of the average and peak strength of the trunk rotator relationship between women and men in the low cable woodchop exercise and the horizontal cable woodchop exercise (**upper panel**) and the average and peak strength of the trunk rotator relationship between the dominant and non-dominant sides in the low cable woodchop exercise and the horizontal cable woodchop exercise (**lower panel**). * Meaningful differences in reliability were identified by a CVratio higher than 1.15.

DS-NDS showed no significant differences in the strengths of the two exercises, except for LCW at $0.50 \text{ m·s}^{-1}$ and HCW exercise with the average strengths in the concentric phase at $0.40 \text{ m·s}^{-1}$ and $0.60 \text{ m·s}^{-1}$ ($p < 0.05$), respectively. Very large to extremely large correlations were observed between DS-NDS in the strength of two TRE ($r = 0.71–0.91$), with significant correlations ($p = 0.001$) (Table 5).

**Table 5.** Correlation (r) and significant differences ($p$ value) for dominant and non-dominant sides on two trunk rotator strengths.

| | | **Pearson Correlation (r)** | ***p*-Value** |
|---|---|---|---|
| **LCW** | | | |
| | $0.50 \text{ m·s}^{-1}$ concentric ND vs. D | 0.87 | 0.76 |
| | $0.50 \text{ m·s}^{-1}$ eccentric ND vs. D | 0.75 | 0.29 |
| Peak strength | $0.70 \text{ m·s}^{-1}$ concentric ND vs. D | 0.79 | 0.40 |
| | $0.70 \text{ m·s}^{-1}$ eccentric ND vs. D | 0.81 | 0.68 |
| | Isometric ND vs. D | 0.90 | 0.36 |
| | $0.50 \text{ m·s}^{-1}$ concentric ND vs. D | 0.91 | 0.03 |
| | $0.50 \text{ m·s}^{-1}$ eccentric ND vs. D | 0.88 | 0.73 |
| Average strength | $0.70 \text{ m·s}^{-1}$ concentric ND vs. D | 0.89 | 0.14 |
| | $0.70 \text{ m·s}^{-1}$ eccentric ND vs. D | 0.90 | 0.57 |
| | Isometric ND vs. D | 0.91 | 0.83 |

**Table 5.** *Cont.*

| | | Pearson Correlation (r) | *p*-Value |
|---|---|---|---|
| **HCW** | | | |
| Peak strength | 0.40 m·s$^{-1}$ concentric ND vs. D | 0.71 | 0.83 |
| | 0.40 m·s$^{-1}$ eccentric ND vs. D | 0.82 | 0.31 |
| | 0.60 m·s$^{-1}$ concentric ND vs. D | 0.78 | 0.37 |
| | 0.60 m·s$^{-1}$ eccentric ND vs. D | 0.90 | 0.69 |
| | Isometric ND vs. D | 0.89 | 0.26 |
| Average strength | 0.40 m·s$^{-1}$ concentric ND vs. D | 0.87 | 0.01 |
| | 0.40 m·s$^{-1}$ eccentric ND vs. D | 0.89 | 0.09 |
| | 0.60 m·s$^{-1}$ concentric ND vs. D | 0.86 | 0.01 |
| | 0.60 m·s$^{-1}$ eccentric ND vs. D | 0.89 | 0.05 |
| | Isometric ND vs. D | 0.89 | 0.08 |

LCW = low cable woodchop; HCW = horizontal cable woodchop; ISO = isometric contraction; ND = non-dominant; D = dominant.

## 4. Discussion

The purposes of this study were to analyze the influence of sex on the reliability of two TRE, compare the reliability of the DS-NDS in the assessment of two TRE, and study the relationship between the DS-NDS and the strength of two TREs. The results indicated that sex affected reliability in the HCW exercise but not in the LCW exercise. The DS-NDS did not affect the reliability of the measures in the HCW exercise. They did so in the LCW exercise, regardless of which manifestation of strength was measured. Considering sex and the dominant side, there were no differences in the HCW exercise, but differences were found in the LCW exercise. In females, the dominant side was more reliable when average strength was taken, and in males, the non-dominant side was more reliable when peak strength was taken. The strength in both exercises did not show significant differences in DS-NDS in almost all conditions. Furthermore, the strong correlations of DS-NDS were very large and extremely large.

Few studies have evaluated STRM with a similar method. Zemková et al. (2017) obtained high reliability (ICC = 0.93–0.97) using a very similar protocol with a standing cable wood chop exercise and weight stack machine with incremental loads (20 to 55 Kg) [15]. Palmer et al. found high reliability (ICC = 0.83–0.98) when evaluating the STRM with the chop and lift test in a half-kneeling position and using a dynamometer and a pulley with 12% and 15% of the individual's body mass for the lift and chop, respectively [14]. However, no study assessed the STRM in isokinetic mode and in a functional way [7] as has been performed in the present study, which is the most novel. The most commonly used devices to assess STRM are isokinetic devices, load cells, and handheld dynamometers. Trunk strength assessments with isokinetic devices have high reliability, but their high cost and their very analytical evaluation of the muscle make them not transferable to daily activities or sporting gestures [27,28]. The trunk assessment with FEMD makes it possible to assess trunk reliability more functionally and closer to sporting gestures. Moreover, the advantage of FEMD over other devices is its ease of use, transportability, and low cost compared to isokinetic classic devices.

The STRM influences daily activities and performances such as golf or kayaking [3,17]. In addition, the STRM is essential to provide stability to the spine, so knowing the reliability of the measurements helps both those in the clinical/medical field and coaches or physical trainers. A difference in the STRM reliability was found in HCW exercise, with females being more reliable. A study that assessed trunk rotator power analyzed the reliability by sex and found no differences, with ICCs ranging from 0.93 to 0.97 [11]. Nevertheless, Keller et al. reported that males are more reliable than females when assessing trunk extensor strength and strength resistance [10]. This may suggest that sex may affect the reliability of trunk tests depending on which exercise is performed. In this case, no sex

differences were found in reliability for the LWC exercise, although the lowest CV was found in the HCW exercise.

On the other hand, the dominant side does not affect the reliability of the measurements in the HCW exercise, but it does in the LCW exercise. Furthermore, depending on which expression of strength is used, DS-NDS is more reliable. On the contrary, in previous studies in which the mean power between DS-NDS has been analyzed, all the ICCs have been around 0.90 [15]. Another study did not consider side laterality, assessing all subjects only on the right side [11]. Knowing the reliability of DS-NDS is very important for coaches and sports physicians. Sometimes the reliability of both sides is not measured. For example, when a training or rehabilitation program is applied, an improvement is sought according to the dominant side without considering the non-dominant side's value. In this case, if the HCW exercise is used, the participant's assessment is not influenced by the use of the dominant or non-dominant side; however, if we use the LCW exercise, both the dominant side and the expression of the measured strength should be taken into account. In addition, the muscular asymmetries that occur in sports such as basketball, football, and volleyball should be considered, where one side of the body is used more than the other, which can cause injuries [1,29]. The subjects evaluated showed no significant differences in the STRM between DS-NDS in most of the conditions assessed.

In this study, the STRM was assessed functionally and in isokinetic mode, which is a novelty for evaluating this musculature. In addition, the strength was compared between DS-NDSs, and the differences in reliability according to sex were studied. However, this study had some limitations. The type of sport performed by the participants was not controlled, and only healthy people were assessed, so the data could not be extrapolated to other subjects. In addition, in some of the LCW exercise evaluation conditions, learning was observed between the test and the retest, so further familiarization with the LCW exercise is recommended. Moreover, when LCW is performed at high speeds, the CVs increase, and statistically significant differences are observed. It can be explained because, despite using functional movement patterns, the isokinetic evaluation situates the subject in an unexplored condition of the movement, that is, moving at a constant speed, in this case, at a higher speed ($0.70 \text{ m} \cdot \text{s}^{-1}$). Therefore, it could influence the CV, reinforcing the idea that when evaluating the STRM, a longer familiarization with the test should be considered.

## 5. Conclusions

The reliability of the assessment of the STRM is not affected by the manifestation of the strength analyzed, whether it is average or peak strength. Furthermore, whether DS-NDS is used to assess the TRM in the HCW exercise is irrelevant. Still, it does affect the LCW exercise, being more reliable on the dominant side. In general, females have higher reliability in assessing the STRM strength in the HCW exercise, with no differences in strength between DS-NDS except for two assessment conditions. In addition, the correlations between DS-NDS were very large or extremely large.

## 6. Practical Applications

These exercises assess the trunk musculature with a similar gesture to those used in sports and provide reliable data. This test is handy for physical trainers or coaches to know the strength of the trunk rotators of their athletes. It is recommended to use the HCW exercise regardless of the athlete's sex. Furthermore, in the case of evaluating only one side, it is recommended to assess the dominant side because it does not show differences in the HCW exercise and is more reliable than the non-dominant side in the LCW exercise. In addition, with these evaluations, it is possible to know if there are muscular imbalances between the dominant and non-dominant sides, which are fundamental in unilateral sports, to avoid injuries and improve the athletes' performance. Based on this study's results, we believe there is a novel and reliable way of assessing trunk rotators.

On the other hand, as the test is an isokinetic test and strength values are obtained, it is possible to determine concentric and eccentric training loads for these exercises to improve the strength of musculature.

**Author Contributions:** Conceptualization, A.R.-P., D.J.-M. and W.R.-F.; methodology, A.R.-P., W.R.-F., M.D.M.-A. and D.M.-G.; formal analysis, A.R.-P., D.J.-M. and D.M.-G.; investigation, A.R.-P., D.J.-M. and D.M.-G.; resources, L.J.C.-R.; data curation, A.R.-P., W.R.-F., M.D.M.-A. and D.M.-G.; writing—original draft preparation, A.R.-P., M.D.M.-A. and W.R.-F.; writing—review and editing A.R.-P. and W.R.-F.; visualization, D.J.-M., L.J.C.-R. and I.J.C.-R.; supervision, D.J.-M., L.J.C.-R. and I.J.C.-R.; project administration, A.R.-P. and W.R.-F.; funding acquisition, L.J.C.-R. and W.R.-F. All authors have read and agreed to the published version of the manuscript.

**Funding:** This research was funded partially by FEDER/Junta de Andalucía- Consejería de Transformación Económica, Industria, Conocimiento y Universidades/Proyecto B-CTS-184-UGR20. The researchers, Angela Rodríguez-Perea, Darío Martínez-García, and Daniel Jerez-Mayorga, have a post-doctoral contract through the program "Recualificación del profesorado universitario. Modalidad Margarita Salas", Universidad de Granada/Ministerio de Universidades y fondos Next Generation de la Unión Europea.

**Institutional Review Board Statement:** The study was conducted in accordance with the Declaration of Helsinki and approved by the Institutional Review Board of the University of Granada (nº 2560/CEIH/2022).

**Informed Consent Statement:** Informed consent was obtained from all subjects involved in the study.

**Data Availability Statement:** All relevant data are within the text.

**Conflicts of Interest:** The authors declare no conflict of interest.

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
