# Peer review of "Influence of Sex and Dominant Side on the Reliability of Two Trunk Rotator Exercises"

_applsci, doi:10.3390/app13042441_

Round 1
Reviewer 1 Report
The study intends to evaluate stability of the STRM assessment and concludes that the manifestation of strength analysed, average or peak strength is not affected. In addition, it also reveals how DS-NDS difference won`t influence assessing the TRM in HCW exercise. However, the LCW exercise is more reliable only when using the dominant side. This study helps us to evaluate the existing muscular imbalances between dominant and non-dominant sides, common in unilateral sports, which further helps athletes to avoid injuries and improve their performance. The authors did a good job.
Minor error
Line 312
In this case, if the HCW exercise is used, participants improvement is not influence by the use of the CV of the dominant or non-dominant side; however, if we use the LCW exercise, both the dominant side and the expression of the measured strength should be taken into account.
This paragraph is little confusing so please rephrase it, specifically from “…participants improvement is not influence by the use of the CV of the dominant or non-dominant side”.
Author Response
Influence of sex and dominant side on the reliability of two trunk rotator exercises.
Dear reviewers,
thank you very much for your time and dedication in reviewing the manuscript. We believe that these comments will improve the quality of our manuscript. Each comment has been specifically addressed and changes to the manuscript have been highlighted in yellow.
Reviewer 1
Comment: The study intends to evaluate the stability of the STRM assessment and concludes that the manifestation of strength analyzed, average or peak strength is not affected. In addition, it also reveals how DS-NDS difference won`t influence assessing the TRM in HCW exercise. However, the LCW exercise is more reliable only when using the dominant side. This study helps us to evaluate the existing muscular imbalances between dominant and non-dominant sides, common in unilateral sports, which further helps athletes to avoid injuries and improve their performance. The authors did a good job.
Answer: Dear reviewer, thank you very much for your comments.
Minor error
Comment: Line 312. In this case, if the HCW exercise is used, participants' improvement is not influence by the use of the CV of the dominant or non-dominant side; however, if we use the LCW exercise, both the dominant side and the expression of the measured strength should be taken into account.
This paragraph is little confusing so please rephrase it, specifically from “…participants improvement is not influence by the use of the CV of the dominant or non-dominant side”.
Answer: Thank you for your comment. The phrase has been rewritten and now it reads: “In this case, if the HCW exercise is used, the participant’s assessment is not influenced by the use of the dominant or non-dominant side; however if we use the LCW exercise, both the dominant side and the expression of the measured strength should be taken into account.”
Reviewer 2 Report
I would like to congratulate the authors on their work.
The paper is a novel, well-researched study on the reliability of trunk rotation exercises.
The manuscript fills a research gap and adequately responds to the proposed objectives. It is also well structured and written.
Therefore, it can be accepted for publication.
Author Response
Influence of sex and dominant side on the reliability of two trunk rotator exercises.
Dear reviewers,
thank you very much for your time and dedication in reviewing the manuscript. We believe that these comments will improve the quality of our manuscript. Each comment has been specifically addressed and changes to the manuscript have been highlighted in yellow.
Reviewer 2
Comment: I would like to congratulate the authors on their work.
The paper is a novel, well-researched study on the reliability of trunk rotation exercises.
The manuscript fills a research gap and adequately responds to the proposed objectives. It is also well structured and written.
Therefore, it can be accepted for publication.
Answer: Dear reviewer, thank you very much for your comments.
Reviewer 3 Report
Congratulations on the work, it is certainly interesting and presents a breakthrough in training improvement.
As it has been confirmed that they are physically active.
Explain the characteristics of the evaluators.
Revise the description of figure two and the images.
Include some reference to the reliability criteria for the Cvratio variable.
Figure 3 is not fully visible. it would be appropriate to explain what values were selected in figure three in the two figures (whether peak and mean force from either side or from one side only).
I think it would be necessary to explain in more depth the Cv values of the different exercises, as they are high and have not been given attention in the discussion.
Some difference is appreciated in the analysis of the results with respect to the speed of execution.
Author Response
Influence of sex and dominant side on the reliability of two trunk rotator exercises.
Dear reviewers,
thank you very much for your time and dedication in reviewing the manuscript. We believe that these comments will improve the quality of our manuscript. Each comment has been specifically addressed and changes to the manuscript have been highlighted in yellow.
Reviewer 3
Congratulations on the work, it is certainly interesting and presents a breakthrough in training improvement.
Answer: Dear reviewer, thank you very much for your comments.
Comment: It has been confirmed that they are physically active.
Answer: Participants were recruited if they had more than 6 months of structured resistance training experience and trained at least 3 days per week. The phrase has been completed with additional information and now it reads: “physically active college students engaged in training sessions at least 3 times a week, with a minimum of 6 months of structured resistance training and without any musculoskeletal injuries.”
Comment: Explain the characteristics of the evaluators.
Answer: Thanks to the reviewer for this comment. We have added the requested information.
Comment: Revise the description of figure two and the images.
Answer: We apologize for this error. The figures have been changed in order and now it reads: “Figure 1. Assessment of Horizontal Cable Woodchop exercise using a functional electromechanical dynamometer.” And “Figure 2. Assessment of Low Cable Woodchop exercise using a functional electromechanical dynamometer.”
Comment: Include some reference to the reliability criteria for the Cvratio variable.
Answer: Three references have been added (23-25).
Garcia-Ramos, A.; Janicijevic, D. Potential Benefits of Multicenter Reliability Studies in Sports Science: A Practical Guide for Its Implementation. Isokinet Exerc Sci 2020, 28, 199–204.
Fulton, S.K.; Pyne, D.; Hopkins, W.; Burkett, B. Variability and Progression in Competitive Performance of Paralympic Swimmers. https://doi.org/10.1080/02640410802641418 2009, 27, 535–539, doi:10.1080/02640410802641418.
Hopkins, W.G.; Hewson, D.J. Variability of Competitive Performance of Distance Runners. Med Sci Sports Exerc 2001, 33, 1588–1592, doi:10.1097/00005768-200109000-00023.
Comment: Figure 3 is not fully visible. it would be appropriate to explain what values were selected in figure three in the two figures (whether peak and mean force from either side or one side only).
Answer: We apologize for this error. In addition, we have improved the layout of the image in the text. The mean value of dominant and non-dominant was calculated for all assessment conditions (upper panel) and the mean value for males and females (lower panel). We have added information to explain it “For the CVratio study, the mean value of dominant and non-dominant side in all assessment conditions was calculated to compare the reliability of males and females and the mean value of males and females to compare the reliability of the dominant and non-dominant side.”
Comment: I think it would be necessary to explain in more depth the Cv values of the different exercises, as they are high and have not been given attention in the discussion.
Answer: We thank the reviewer for his comment. The requested information has been added to the discussion. It now reads: “Moreover, when LCW is performed at high speeds, the CVs increase, and statistically significant differences are observed. It can be explained because despite using functional movement patterns, the isokinetic evaluation situates the subject in an unexplored condition of the movement, that is, moving at a constant speed, in this case, at a higher speed (0.70m·s-1). Therefore, it could influence the CV, reinforcing the idea that when evaluating the STRM, a longer familiarization with the test should be considered.”
Comment: Some difference is appreciated in the analysis of the results with respect to the speed of execution.
Answer: Thanks for your comment, but previous literature showed that higher isokinetic velocities have been associated with higher measurement error, i.e., lower reproducibility [1].
- Caruso, J.F.; Brown, L.E.; Tufano, J.J. The Reproducibility of Isokinetic Dynamometry Data. Isokinetics and Exercise Science 20 2012, 20, 239–253, doi:10.3233/IES-2012-0477.